# Prospective cohort study to investigate the burden and transmission of acute gastroenteritis in care homes: epidemiological results

Thomas Inns ![ORCID] ,[1,2] Anna Pulawska-Czub,[2,3] John P Harris,[1,2] Roberto Vivancos,[2,4] Nicholas J Beeching,[2,5,6] Miren Iturriza-Gomara,[2,3] Sarah J O'Brien[2,7]

For numbered affiliations see end of article.

**Correspondence to**
Thomas Inns;
thomas.inns@liverpool.ac.uk

## ABSTRACT

**Objectives** To estimate the incidence of gastroenteritis in individuals in care homes.

**Design** Prospective cohort study.

**Setting** Five participating care homes in North West England, UK.

**Participants** Residents and staff present at the five study care homes between 15 August 2017 and 30 May 2019 (n=268).

**Outcome measures** We calculated incidence rates for all gastroenteritis cases per 1000 person-years at risk and per 1000 bed-days at risk. We also calculated the incidence rate of gastroenteritis outbreaks per 100 care homes per year.

**Results** In total 45 cases were reported during the surveillance period, equating to 133.7 cases per 1000 person-years at risk. In residents the incidence rate was 0.62 cases per 1000 bed-days. We observed seven outbreaks in all care homes included in surveillance, a rate of 76.4 outbreaks per 100 care homes per year. 15 stool samples were tested; three were positive for norovirus, no other pathogens were detected.

**Conclusions** We found that surveillance of infectious gastroenteritis disease in care homes based on outbreaks, the current general approach, detected a majority of cases of gastroenteritis. However, if policymakers are to estimate the burden of infectious gastroenteritis in this setting using only routine outbreak surveillance data and not accounting for non-outbreak cases, this study implies that the total burden will be underestimated.

## INTRODUCTION

Gastrointestinal (GI) infections are an important issue in care homes for the elderly (also known as long-term care facilities). Care home residents are more susceptible to infectious gastroenteritis and the environment is ideal for transmission of gastroenteritis.[1] Because infection control measures are challenging to implement, further infections and outbreaks frequently occur based on a single index case.[2] In this population, GI infections can cause more severe morbidity, hospitalisation, and are associated with greater mortality.[3 4]

### Strengths and limitations of this study

► To our knowledge this is the first systematic active surveillance study of gastroenteritis in care home residents in UK.
► Prospective cohort design with active follow-up of individual care home residents by fully trained research nurses.
► Small number of care homes included and so results might not be generalisable.
► Challenges in obtaining stool samples in a timely manner.
► Study period coincided with a low incidence of norovirus in the community.

Surveillance of infectious gastroenteritis in care homes varies in presence and scope in different countries, and where it exists it is focused on the detection of outbreaks. These outbreak surveillance systems exist in countries such as France, Australia and England.[5–7] Using these surveillance data, it is possible to estimate the burden of care home gastroenteritis outbreaks.[8] However this does not account for any sporadic (non-outbreak related) disease.

The incidence of gastroenteritis in care homes is poorly researched, with few studies published over the last 40 years, the majority originating in USA.[9–12] The objective of this study was to estimate the incidence of gastroenteritis in individuals in care homes in North West England; therefore, addressing this gap in the evidence base, and providing data to understand the burden of infectious gastroenteritis in this setting.

## METHODS

The study protocol has been published and the methods are fully described there.[13] Briefly, we conducted a prospective cohort study in residents of five care homes in North

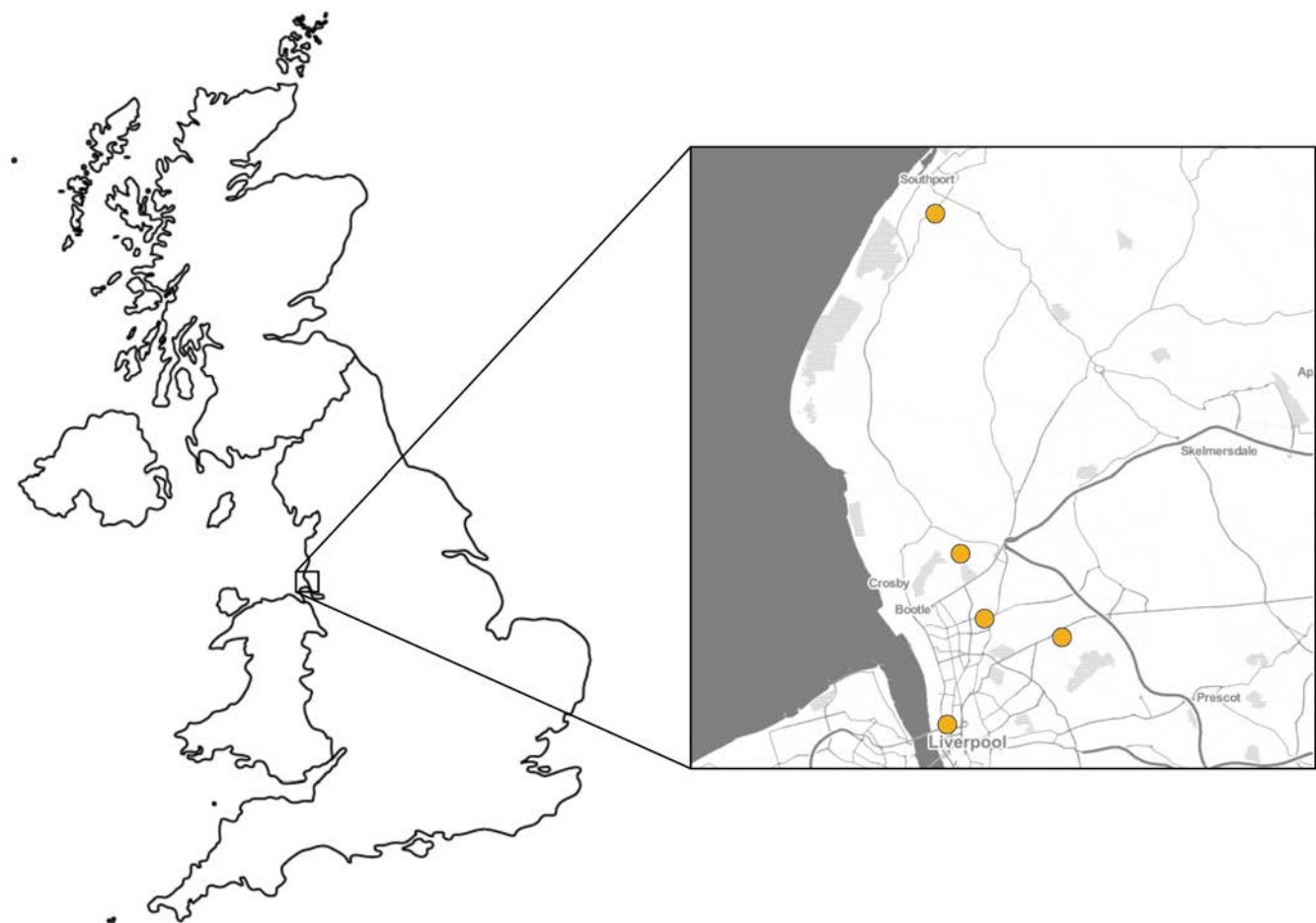

**Figure 1** Location of study sites, England, 2017–2019.

West England. The study took place from 15 August 2017 to 30 May 2019.

## Study population

The sampling frame was the total number of residential care homes for the elderly in the local authorities of Liverpool and Sefton, registered with the Care Quality Commission. The five care homes selected were a convenience sample of care homes in this sampling frame that were approached and agreed to participate. The locations of the study care homes are shown in figure 1. All study care homes were recruited prospectively at the same time; no other care homes were invited to participate and declined. All residents and staff members who were present at study care homes during the study period were eligible to participate. Eligible participants with capacity to consent were consented by study research nurses; for those without capacity to consent a nominated person who met the criteria described in Section 32 of the Mental Capacity Act 2005 was asked to provide consent.

## Surveillance system

The number of residents and staffing levels at each care home were collected at the start of the study period using a questionnaire, administered to each care home manager. Data including: age, sex, general practitioner,

date of arrival at the home and position in the home were collected in person by trained research nurses. Participants were recruited between 15 August 2017 and 08 November 2018. Participants were recruited from the start of the study period, with new residents and staff being recruited when entering the care home. No participants were ill with gastroenteritis at the point of recruitment or recruited as a result of such illness. Study research nurses employed active surveillance by visiting each study care home on a weekly basis to ascertain new participants, episodes of illness meeting the case definition and details about participants withdrawing from the study. During these visits, study research nurses met with key leadership staff to understand any changes at the home in the preceding week. For each case, information including onset date, medical history, duration of symptoms, complications and hospitalisation were collected using a questionnaire. Case report questionnaires were completed by a study research nurse.

## Case definitions

The primary outcome was a case of gastroenteritis. Gastroenteritis cases were defined as persons in the study population with vomiting (two or more episodes of vomiting in a 24-hour period) OR diarrhoea (three

or more loose stools in a 24-hour period), OR vomiting AND diarrhoea (one or more episodes of both symptoms in a 24-hour period). Confirmed cases were defined as cases with a positive laboratory diagnosis of an infectious cause. Non-infectious causes such as long-standing diarrhoea associated with disability or incontinence and ingestion of laxative drugs were excluded from the study case definition based on the clinical judgement of a study research nurse. Outbreaks were defined as two or more cases occurring in an institution, with onset of illness being within 5 days.

## Study size

As described in the study protocol, the target study sample size was for 268 participants to be included.[13]

## Microbiological analysis

For each case, participants were asked to provide a faecal sample to determine the cause of symptoms; these samples were collected as soon as possible after onset of illness. Samples were sent to Liverpool Clinical Laboratories, based in the Royal Liverpool University Hospital. Diagnostic tests were conducted in real time and results reported to the study team. Samples were tested for 16 pathogens using Luminex xTAG Gastrointestinal Pathogen Panel (Luminex Molecular Diagnostics, Austin, Texas, USA). Results were reported to the study team and copied to the participant's general practitioner. The operation of this study was designed so that it did not interfere with public health action.

## Statistical methods

We characterised the demographics of study participants and described differences between residents and staff. We described the distribution of gastroenteritis case onset date over time, along with the number and incidence rate of outbreaks (with binomial 95% CI). We calculated incidence rates for all gastroenteritis cases. Participants could contribute multiple illness episodes. The denominator was the person-time at risk (PTAR) in study participants; incidence rates are expressed per 1000 person-years at risk for all groups and per 1000 bed-days for residents. Bed-days were defined as days that the resident was present in the care home; participant PTAR was censored if they left the care home. PTAR was calculated in the same way for residents and staff and commenced when participants

were recruited into the study and was censored when they left the study care home; otherwise it was censored when the surveillance period ended on 30 May 2019.

## Patient and public involvement

Patients, carers or members of the public were not actively involved in the design of this research.

## RESULTS

In total 268 participants (159 residents and 109 staff) were recruited into the study from five care homes. Seventy-nine participants (59 residents and 20 staff) withdrew from the study before the end of the surveillance period. None of these withdrawals were due to serious adverse events. Fifty-five (93%) of resident withdrawals were due to death from an unrelated cause, with four residents leaving the care home to return to live independently. All 20 staff withdrawals were due to the participant leaving employment at the study care home. The participants contributed a total of 122 898 days PTAR (66 489 days PTAR for residents; 56 409 days PTAR for staff). The median contribution of PTAR was 504 days (range 2–837 days). A summary of participant demographics is shown in table 1. The median age of participants was 71 years (range 19–99); the median age of residents was 82 and the median age of staff was 44. In total, 190 participants were female (70.9%); 62.9% of residents and 82.6% of staff were female. It was not possible to calculate the participation rate as the denominator of staff and residents in each home was not available.

In total 45 cases of gastroenteritis were reported during the surveillance period, equating to 133.7 cases per 1000 person-years at risk. The incidence rate of illness in residents was 225.2 cases per 1000 person-years at risk and the incidence rate of illness in staff was 25.9 cases per 1000 person-years at risk (table 2). For residents, the incidence rate was 0.62 cases per 1000 bed-days. Two participants became a case twice during the study. No cases were excluded based on a non-infectious cause of diarrhoea.

The distribution of case onset dates is shown in figure 2. A majority of cases were reported in September and October during both winters. We observed seven outbreaks in study participants in these care homes, an incidence rate of 76.4 outbreaks per 100 care homes per

**Table 1** Demographics of study participants, by care home and role in the home

| Care home | Total | | | Residents | | | Staff | | |
|---|---|---|---|---|---|---|---|---|---|
| | N | Median age | % Female | N | Median age | % Female | N | Median age | % Female |
| 1 | 88 | 79 | 59 | 69 | 82 | 58 | 19 | 37 | 63 |
| 2 | 45 | 79 | 62 | 34 | 85 | 62 | 11 | 55 | 64 |
| 3 | 80 | 55 | 83 | 33 | 78 | 70 | 47 | 44 | 92 |
| 4 | 29 | 59 | 79 | 13 | 86 | 69 | 16 | 43 | 88 |
| 5 | 26 | 59 | 81 | 10 | 88 | 70 | 16 | 49 | 88 |
| Total | 268 | 70 | 71 | 159 | 82 | 63 | 109 | 44 | 83 |

**Table 2** Case incidence rates, by care home and role in the home

| | Total | | | Residents | | | Staff | | |
|---|---|---|---|---|---|---|---|---|---|
| Care home | PTAR (years) | Cases | Incidence rate (1000 person-years) | PTAR (years) | Cases | Incidence rate (1000 person-years) | PTAR (years) | Cases | Incidence rate (1000 person-years) |
| 1 | 110.2 | 15 | 136.1 | 80.8 | 15 | 185.6 | 29.4 | 0 | 0 |
| 2 | 55.9 | 6 | 107.3 | 37.7 | 6 | 159.3 | 18.3 | 0 | 0 |
| 3 | 108.3 | 16 | 147.8 | 38.7 | 13 | 335.5 | 69.5 | 3 | 43.1 |
| 4 | 35.9 | 6 | 167.1 | 14.8 | 5 | 337.4 | 21.1 | 1 | 47.4 |
| 5 | 26.2 | 2 | 76.5 | 10.0 | 2 | 200.3 | 16.2 | 0 | 0 |
| Total | 336.5 | 45 | 133.7 | 182.0 | 41 | 225.2 | 154.4 | 4 | 25.9 |

year (95% CI: 44.2–92.9 outbreaks per 100 care homes per year). Three outbreaks were observed in care home 3 (five, six and three cases, respectively), two outbreaks were observed in care home 1 (eight cases and seven cases) and one outbreak was observed in both care homes 2 (five cases) and 4 (six cases). No outbreaks occurred in care home 5 during the study. In total, 40 (89%) cases were defined as part of an outbreak. The most frequently reported symptoms were: diarrhoea (62%), vomiting (47%), nausea (22%) and abdominal pain (6%). No cases reported bloody stool, fever or headache. Seven cases (16%) reported both diarrhoea and vomiting. Duration of illness for cases was not available.

At least one faecal sample was collected for 15 cases (33.3%) of the 45 reported cases. No samples were collected for any of the four cases in staff. The 15 samples were tested for multiple pathogens. Norovirus was detected in three samples. No pathogen was detected in 12 samples.

For the 15 stool specimens which were received, the median time delay between onset of symptoms and the

sample being taken was 3 days (range 0–18 days). The median delay for samples positive for norovirus was 0 days (range 0–1 days). This was significantly shorter (Wilcoxon rank sum test, p value=0.016) than the delay for samples which were negative (median 4 days, range 1–18 days).

## DISCUSSION
### Main findings
In this active surveillance study using a prospective cohort design we recorded gastroenteritis cases in care homes over a 22-month period and observed seven outbreaks in study participants, a rate of 76.4 outbreaks per 100 care homes per year. Both this point estimate and the lower bound of the 95% CI are greater than the incidence rate of 37.1 outbreaks per 100 care homes per year reported during routine, passive surveillance in the same geographical area between 2012 and 2016.[7] This difference may reflect increased reporting of illness due to regular

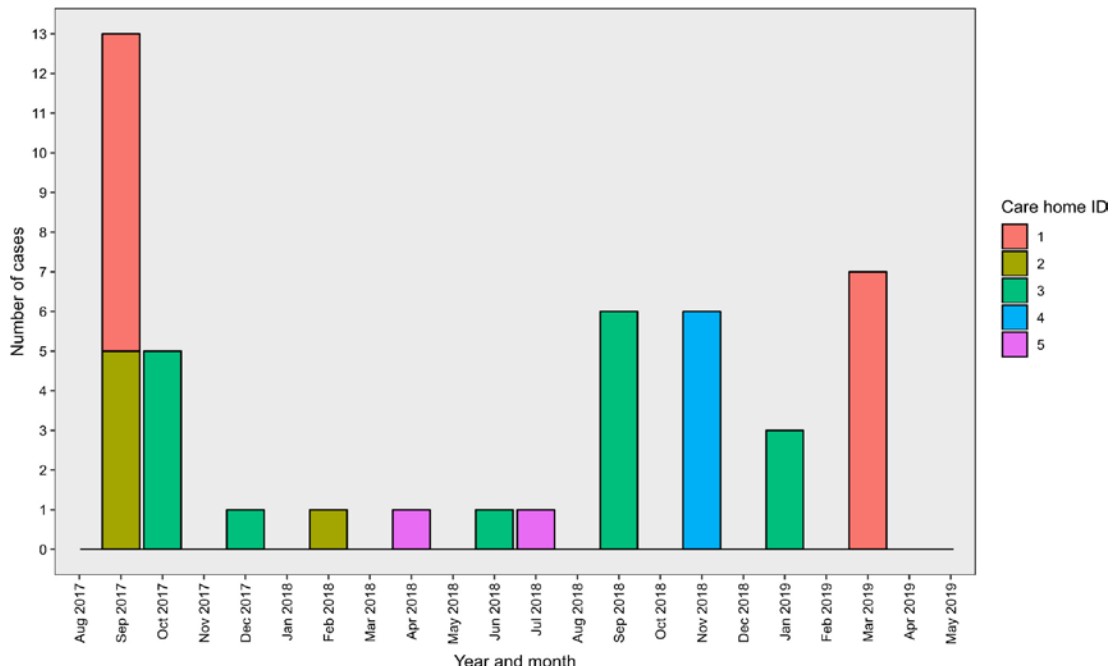

**Figure 2** Epidemic curve showing distribution of cases by month and study care home.

contact with the care homes as part of the study, which is likely to have improved ascertainment of outbreaks.

We found that the incident rate of illness in participants was 133.7 per 1000 person-years at risk, and that the rate was far higher in residents (225.2 per 1000 person-years) than in staff (25.9 per 1000 person-years). This difference could be caused by a number of factors: it may reflect trends in the wider community where norovirus incidence is higher in older people than those of working age,[14] good hygiene and infection control practices by staff, reduced exposure in staff who go home when not on shift, the increased susceptibility of elderly residents who are physically debilitated,[15] and illness not being reported by staff, some of whom do not receive sick pay. The incidence rate of illness in residents can also be expressed as 0.62 cases per 1000 bed-days; this study is the first time this metric has been estimated for care homes in UK and as such will provide data to inform any modelling of the economic burden of gastroenteritis in this setting.

In this study, we observed that 89% of cases were defined as part of an outbreak. This comparatively low level of individual cases may be due to factors such as; the susceptible nature of residents, the high degree of potential contacts and the difficulty of maintaining hygiene. These factors could explain why people in a care home who acquire a GI infection are likely to infect another and therefore GI illness in these settings frequently causes outbreaks. This finding therefore supports the continued surveillance of GI disease in care homes being focused on outbreaks as this constitutes the majority of disease burden.

The study protocol was for a stool sample to be submitted for each case; in practice this only occurred for 33% of cases. Of the 15 samples tested, norovirus was the only pathogen identified, being found in three cases. Despite being tested for, no other pathogens were identified, which may have been associated with delay between symptom onset and stool submission. Due to the small number of stool samples in this study, caution should be exercised if these results are to be used to infer the proportion of gastroenteritis in care homes caused by norovirus.

### Strengths

One of the key strengths of this study was its active surveillance design, whereby a research nurse visited each study site each week to check on the status of study participants. During the 22-month duration of the study, this was a resource-intensive approach and meant that care homes involved in the study were constantly aware of the need to report illness in study participants. This active surveillance design meant that our study is likely to have recorded a higher proportion of cases than an alternative passive surveillance design, an assertion supported by the incidence rate being higher than that reported from the same area during routine surveillance.

This is the first active surveillance study to follow-up individuals in a care home setting for GI illness. The advantage of this study design is that the individual level of participation and surveillance allowed the calculation of PTAR and the recording of sporadic cases of illness, in addition to outbreaks. This is a valuable addition to the literature as the description of individual cases, including sporadic illness, is not covered in other studies that mainly focus on the burden of gastroenteritis outbreaks. These findings are key to understanding the burden of sporadic gastroenteritis in care homes, which is important when calculating the total burden of illness in this setting.

An additional strength of this study was the capacity to test each of the cases for a wide variety of pathogens. In contrast to other studies which focus on testing for norovirus or other viral pathogens in care home settings, we used a multiplex PCR test which was capable of detecting 15 pathogens. By using the Luminex Gastrointestinal Pathogen Panel, we were confident that we had coverage for the most likely known pathogens and would be able to detect them in any cases that arose during the study.

### Limitations

A key limitation of this study was that it included a small convenience sample of care homes in one area of England. Due to the nature of the study, it was only possible to include those care homes which were approached and agreed to participate. It may have been that the five care homes included in the study varied systematically from the others in the sampling frame in aspects such as: the level of care provided, the vulnerability of residents to infection, the socioeconomic status of residents and infection prevention and control practices. However, it was not possible to obtain such information on all homes in the sampling frame and therefore it is not possible to make a formal comparison. Due to the resource-intensive active surveillance design it was only possibly to include a maximum of five sites in this study. It may be that the small number of geographically clustered care homes in this study limits the generalisability of these findings to other areas of the country and internationally. The inferences that can be made from this study may also be affected by the duration of the surveillance period; although the 22 months of the study include two winters, it may have been that the circulating viruses during these seasons were atypical.

Another potential limitation may have been that the participants in our study care homes who consented to take part were systematically different from those in the care homes who did not take part. The consenting process to enrol participants in this study was agreed with the relevant ethics committee and meant that the study team did not have access to the personal information of staff or residents at the home who did not consent to take part. Therefore, it was not possible to compare the characteristics of those who took part to those who did not. Furthermore, by following the agreed consenting process, because we could not record departures and arrivals of persons at the home who were not participants, although we knew the capacity of each home, we

could not calculate the participation rate in each home. Although it was not possible to formally calculate the participation rate, it is possible to note that participation could have been higher. One reason for this was the consenting process for those (mainly elderly) residents without capacity to consent. Safeguarding the rights of such people is very important, but the process we were asked to follow made it very difficult to identify and contact the correct person to represent the interests of that person. Therefore, fewer residents without capacity were enrolled in the study than would have otherwise been the case.

One issue that has previously been identified when studying gastroenteritis illness in care homes is the difficulty in obtaining stool samples for pathogen testing.[7] Even with weekly visits to the care homes, we only obtained stool samples from 33% of the cases. For the samples we received, we found that frequently these were taken several days after the onset of symptoms and this may account for the 80% of samples where no pathogen was identified. During the study we acknowledged this difficulty in obtaining stool samples and implemented a £5 voucher scheme on 28 June 2018 to incentivise stool collection. Unfortunately, this was not particularly effective as 30% of cases submitted a stool sample before this point, compared with 36% afterwards. This low proportion of stool samples shows one of the challenges of operating the study in very busy care home environments with staff working at a level where they do not have much excess capacity.

### Results in the context of the international literature

In this study, the incidence rate of infectious gastroenteritis in care home residents was estimated to be 0.62 cases per 1000 bed-days. This finding is substantially higher than the mean global incidence estimate in a systematic review of published surveillance; the pooled estimate of incidence from this meta-analysis was 0.40 (95% CI 0.27–0.56) episodes per 1000 bed-days.[9] However there was considerable heterogeneity between the 15 studies, with the highest incidence (1.9 episodes per 1000 bed-days) being reported from a German study using electronic health records.[16] The authors of this systematic review were surprised with the low rate of gastroenteritis in the meta-analysis and the results of our study support this observation, being a substantially higher incidence. This higher incidence is likely to reflect enhanced case-finding in our study due to the active surveillance design. However, the incidence rate from our study was still lower than that reported in persons aged over 65 years living in the community.[17]

### CONCLUSION

The key implication for policymakers to be drawn from this study is that we found that surveillance of infectious gastroenteritis disease based on outbreaks in care homes, the current general approach, detected a majority of cases of gastroenteritis. However, if policymakers are to estimate the burden of infectious gastroenteritis in this setting using only routine outbreak surveillance data and not accounting for non-outbreak cases, this study implies that the total burden will be underestimated. Combining findings from this study with data on the distribution of outbreaks in care homes would be a way for future research to fully estimate the burden of infectious gastroenteritis in this setting.

**Author affiliations**
[1]Institute of Population Health Sciences, University of Liverpool, Liverpool, UK
[2]NIHR Health Protection Research Unit in Gastrointestinal Infections, University of Liverpool, Liverpool, UK
[3]Institute of Infection and Global Health, University of Liverpool, Liverpool, UK
[4]Field Epidemiology Services, Public Health England, London, UK
[5]Tropical and Infectious Disease Unit, Royal Liverpool University Hospital, Liverpool, UK
[6]Clinical Sciences Group, Liverpool School of Tropical Medicine, Liverpool, UK
[7]School of Natural and Environmental Science, Newcastle University, Newcastle upon Tyne, UK

**Acknowledgements** The authors would like to acknowledge the hard work and dedication of the research nurses from Royal Liverpool and Broadgreen University Hospitals NHS Trust and North West Coast Clinical Research Network who worked on this study. We would like to thank the management and staff at each of the study care homes for their engagement in participation in this study. The authors would also like to thank Jonathan M Read and David J Allen for their contribution to the study protocol.

**Contributors** TI, AP-C, JPH, RV, NJB, MI-G and SJO conceived and designed the study. TI and AP-C co-ordinated data collection. TI undertook the analysis, wrote the first draft and revised the manuscript. TI, AP-C, JPH, RV, NJB, MI-G and SJO provided input to the manuscript drafting process. All authors reviewed and approved the final manuscript.

**Funding** This research was funded by the National Institute for Health Research Health Protection Research Unit (NIHR HPRU) in Gastrointestinal Infections at University of Liverpool in partnership with Public Health England (PHE), in collaboration with University of East Anglia, University of Oxford and the Quadram Institute. Thomas Inns is based at the University of Liverpool. The views expressed are those of the author(s) and not necessarily those of the NHS, the NIHR, the Department of Health or Public Health England.

**Map disclaimer** The depiction of boundaries on this map does not imply the expression of any opinion whatsoever on the part of BMJ (or any member of its group) concerning the legal status of any country, territory, jurisdiction or area or of its authorities. This map is provided without any warranty of any kind, either express or implied.

**Competing interests** None declared.

**Patient consent for publication** Not required.

**Ethics approval** The study was approved by the North West–Greater Manchester South NHS Research Ethics Committee (REC Reference: 16/NW/0541).

**Provenance and peer review** Not commissioned; externally peer reviewed.

**Data availability statement** Data are available upon reasonable request.

**ORCID iD**
Thomas Inns http://orcid.org/0000-0002-1218-958X

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
