## [Reviewer comments · BMJ Open]

ARTICLE DETAILS

TITLE (PROVISIONAL)	Prospective cohort study to investigate the burden and transmission of acute gastroenteritis in care homes: epidemiological results
AUTHORS	Inns, Thomas; Pulawska-Czub, Anna; Harris, John; Vivancos, Roberto; Beeching, Nicholas; Iturriza-Gomara, Miren; O'Brien, Sarah

VERSION 1 - REVIEW

REVIEWER	Cristina Cardemil CDC, USA
REVIEW RETURNED	30-Sep-2019

GENERAL COMMENTS	General: This is a prospective cohort study in care homes (e.g., long term care facilities) in northwest England to examine the burden of acute gastroenteritis (AGE). The authors conducted surveillance from 15 August 2017 to 30 May 2019 in a convenience sample of 5 care homes to better define the burden of AGE. At first read, the study appears to be well designed; however more details are needed to fully understand the methods. The main objective to look at burden is stated; however it is unclear if the authors intend to focus on endemic vs outbreak burden, or even a comparison between the two, as they go back and forth in stating the relative importance of each. The statistical analysis is partially stated upfront but then results and discussion are presented that do not follow the initial description, which results in an unfocused paper. With substantial revisions to the objective statement, abstract, methods, results and discussion, this paper has potential to be a valuable contribution to the literature. Specifics: Abstract: Results--The statement "We observed 7 outbreaks in study participants" is not clear—do the authors mean 7 outbreaks across all care homes included in surveillance in the study time frame? Conclusions-- do not follow the background, methods, and results sections. First sentence of the conclusion section implies data/knowledge that is not presented. Second and last sentence of the conclusion is difficult to interpret from the data presented. Also, the conclusion suggests the authors aim was to compare a new method for measuring incidence, but this idea is not introduced in the background or in the methods, leaving the reader with the
--

impression that the study aim and design were not carefully considered.

Introduction:

There are a number of published studies from the US on AGE outbreaks in long term care facilities, as the authors state; however, no references are included. Suggest including to help the reader understand the specific gap in the literature that the authors are looking to fill (e.g., comparison with endemic disease rates in US or other countries?)

Methods

More information on how active surveillance was employed would be beneficial. The authors state that study research nurses visited each study care home on a weekly basis. Did they utilize systematic methods to ascertain new cases, e.g. 1) ask staff at each care home to maintain a line list of possible cases; 2) meet with key staff (e.g., charge nurses or facility leadership) on a regular basis to identify possible participants for inclusion; 3) review line lists kept by nursing staff of potentially eligible cases with certain regularity; 4) review all laboratory testing results for stool specimens submitted for testing; 5) review all admission diagnoses for possible inclusion in the study; etc.

Results:

-What is the overall participation rate of staff and residents from these 5 care homes? Suggest you provide an overall denominator for staff, and for residents, as well as the proportion that were ultimately included in the final analysis.

-The dropout rate was substantial (59 residents out of 159=38%; 20 staff out of 109=18%). Suggest you provide the reasons for dropout, after the participants had enrolled.

-how did you define bed-days. In the methods, only incidence rates as 1000 person-years at risk are mentioned, and presumably the time the residents were at risk was only when they were in the care homes, as the authors state they were censored when they left the study care home.

-as in the abstract, the statement 'we observed seven outbreaks in study participants in these care homes' is unclear. Suggest you add more information on these outbreaks—which care homes, how many participants were involved in each outbreak, and how the outbreak cases were similar or different from endemic (sporadic) cases (e.g., symptoms, gender, age, location in facility, etc)

-page 7, lines 128-13. The following statement is unclear: "The median time difference for samples 129 positive for norovirus was 0 days (range 0-1 days). This was significantly shorter (Wilcoxon rank sum 130 test, p-value = 0.016) than the difference for samples which were negative (median 4 days, range 1- 131 18 days)" what do the authors mean by time difference? Are they saying that samples that had a faster collection time from the onset of symptoms were more likely to be positive for norovirus? If so, please clarify. Also, this is to be expected as norovirus is most likely to be detected in the first few days after onset of illness, even if it can shed in the stool for weeks following the initial infection.

	Discussion -first paragraph comparing incidence rate in the current study (76.4 outbreaks per 100 care homes per year) with a previous study (37.1 outbreaks/100 care homes/year)—the authors have not included any confidence intervals or measures of uncertainty, so the reader cannot assess if this is a true difference. -first mention that 89% of cases were part of an outbreak. Where are these data in the results? Again, more of a description on the outbreaks (e.g., number of cases in each, facility control measures, etc) and outbreak cases is needed to understand their importance. -page 10, line 252: how did the authors calculate the following data point: “this study implies that the total burden will be underestimated by around 25%.” What do the authors mean by ‘burden’ here—clearly, it is not the % of cases included in outbreaks out of the total, which was previously stated in the discussion as 89%. Additionally, if this is the main finding of the study, warranting inclusion in the concluding paragraph as well as the abstract, it needs to be clear to the reader from the results and prior discussion how this was calculated. -the majority of the discussion focuses on outlining the strengths and weaknesses of the study, rather than orienting the reader to the importance of the study findings.
--	---

REVIEWER	Carly Adams Emory University Rollins School of Public Health, USA
REVIEW RETURNED	22-Oct-2019

GENERAL COMMENTS	The methods described will sufficiently allow the study to be repeated once authors elaborate on how and when participants were recruited (see attached). Review: Prospective cohort study to investigate the burden and transmission of acute gastroenteritis in care homes: epidemiological results This paper presents results from a prospective cohort study examining the incidence of individual gastroenteritis cases in care homes in North West England, United Kingdom. Authors used active surveillance data to calculate the incidence rate of gastroenteritis illness in care homes, both overall and stratified by residents and staff. Because studies examining gastroenteritis incidence in this setting are limited, and surveillance systems focus on gastroenteritis outbreaks rather than individual cases, this paper confronts the important issue of estimating the incidence of gastroenteritis in care homes. Overall, the study is well conducted and uses prospectively collected data, minimizing concerns about reporting bias and/or recall bias. Authors adequately address limitations of the data, noting concerns about generalizability of results to other areas of England and different developed countries and concerns about selection bias due to potential differences between study participants and eligible individuals who did not (or were unable to) consent to take part. However, it is not entirely clear from the paper how PTAR was
---

calculated for participants in the study, leading to concerns about incidence rate calculations. Furthermore, there are discrepancies in the paper for reported incident rates, particularly the incident rate for residents.

Specific comments follow.

Major:

1. Page 4, Lines 39-46: While it states that PTAR commenced upon recruitment into the study, it is unclear when and how participants were recruited. Was there a start date when all participants were recruited, after which new care home residents/staff were recruited upon entering the care home? Was this start date the same for all care homes? I would be particularly concerned if new participants were ascertained because they became ill with gastroenteritis, and were recruited into the study as a result of this illness.
2. Page 4, Lines 39-46: Were participants screened for gastroenteritis prior to recruitment? I recommend addressing this to ensure prevalent cases were not included in the incidence rate calculations.
3. Page 5, Line 94: How was PTAR calculated for staff? Was it calculated the same for staff and residents? Particularly, was the amount of time spent at the care home (i.e., hours worked) taken into account? If not, I recommend addressing this in the limitations.
4. Page 6, Lines 106 and 108; Page 7, Lines 149-150; Page 10, Line 236; Table 2: The incidence rate in care home residents, which is one of your main findings, is not consistent. On page 7, the incidence rate in residents is listed as 0.69 cases per 1000 bed-days. On pages 8 and 11, the incidence rate in residents is listed as 0.78 cases per 1000 bed-days. It is unclear why these are different.

Minor:

5. Page 3, Lines 28-29: While participating care homes could not be compared to those that did not participate, it would be useful to have more information about the care homes that did agree to participate. Were they recruited prospectively or reactively? How many other homes were invited to participate but declined? Were they all recruited at the same time?
6. Page 5, Lines 39-40: Was this questionnaire administered to all care homes at the beginning of the study period? Was it administered only once? Please clarify.
7. Page 4, Lines 53-55: How were non-infectious causes determined? Were these determined by a trained nurse as well? By self-report? It would be useful to include the number of cases excluded for this reason.
8. Page 5, Line 76: Incidence rates were not calculated for norovirus cases only, as only 3 cases tested positive for norovirus. I recommend removing this.
9. Page 6, Line 111 (Table 2):
 1. Please specify units for the incident rate.
 2. Because the incident rate is presented as cases per 1,000 person years, I recommend converting PTAR to years.
 3. PTAR for all cases is listed as 82,358 days, however this column sums to 122,898.
 4. The number of cases for all residents is listed as 41, however this column sums to 39.

	5. The incidence rate for residents is listed as 252.5, however $41/(66,489/365) = 225.1$. Furthermore, $41/66,489 = 0.62$ per 1000 bed-days, which is different than that presented in the text. 10. Page 7, Line 124: Table 2 lists 4 cases in staff (not 3). 11. Page 7, Line 145: Another likely reason for this difference is that staff are not exposed as long as residents. Staff will go home after working (i.e., they are only exposed while working at the care home), whereas residents are presumably always being exposed. 12. Page 10, Lines 244-245: The study referenced here estimated prevalence, not incidence, of gastroenteritis in the community. I recommend removing this, as the incidence rate from this study should not be compared to a prevalence estimate.
--	---

VERSION 1 – AUTHOR RESPONSE

Reviewer: 1

Reviewer Name: Cristina Cardemil

Institution and Country: CDC, USA

Please state any competing interests or state 'None declared': None declared

General:

This is a prospective cohort study in care homes (e.g., long term care facilities) in northwest England to examine the burden of acute gastroenteritis (AGE). The authors conducted surveillance from 15 August 2017 to 30 May 2019 in a convenience sample of 5 care homes to better define the burden of AGE. At first read, the study appears to be well designed; however more details are needed to fully understand the methods. The main objective to look at burden is stated; however it is unclear if the authors intend to focus on endemic vs outbreak burden, or even a comparison between the two, as they go back and forth in stating the relative importance of each. The statistical analysis is partially stated upfront but then results and discussion are presented that do not follow the initial description, which results in an unfocused paper. With substantial revisions to the objective statement, abstract, methods, results and discussion, this paper has potential to be a valuable contribution to the literature.

Specifics:

Abstract:

Results--The statement "We observed 7 outbreaks in study participants" is not clear—do the authors mean 7 outbreaks across all care homes included in surveillance in the study time frame?

We have clarified as requested in the abstract.

Conclusions-- do not follow the background, methods, and results sections. First sentence of the conclusion section implies data/knowledge that is not presented. Second and last sentence of the conclusion is difficult to interpret from the data presented. Also, the conclusion suggests the authors aim was to compare a new method for measuring incidence, but this idea is not introduced in the background or in the methods, leaving the reader with the impression that the study aim and design were not carefully considered.

We have re-worded the conclusion section in the abstract in light of this comment. The conclusion now directly references knowledge presented and makes statements which fully follow results of the analysis. This has also been changed in the conclusion section of the main text (lines 269-274).

Introduction:

There are a number of published studies from the US on AGE outbreaks in long term care facilities, as the authors state; however, no references are included. Suggest including to help the reader understand the specific gap in the literature that the authors are looking to fill (e.g., comparison with endemic disease rates in US or other countries?)

With regards to published studies from the US, we have added two US-specific references to supplement the two already cited (line 16)

Methods

More information on how active surveillance was employed would be beneficial. The authors state that study research nurses visited each study care home on a weekly basis. Did they utilize systematic methods to ascertain new cases, e.g. 1) ask staff at each care home to maintain a line list of possible cases; 2) meet with key staff (e.g., charge nurses or facility leadership) on a regular basis to identify possible participants for inclusion; 3) review line lists kept by nursing staff of potentially eligible cases with certain regularity; 4) review all laboratory testing results for stool specimens submitted for testing; 5) review all admission diagnoses for possible inclusion in the study; etc.

We accept the reviewer's point and have added detail on what the research nurses did at the homes (lines 50-51). Unfortunately the other methods mentioned by the reviewer above were not possible in this study due to ethical concerns.

Results:

-What is the overall participation rate of staff and residents from these 5 care homes? Suggest you provide an overall denominator for staff, and for residents, as well as the proportion that were ultimately included in the final analysis.

We thank the reviewer for this point. In the discussion (lines 233-238) we outlined how it was not possible to calculate a participation rate due to the consenting process agreed with the ethics

committee. To clarify this point, we have amended the wording in the discussion (lines 236-237) and added text to the results section (lines 111-112).

-The dropout rate was substantial (59 residents out of 159=38%; 20 staff out of 109=18%). Suggest you provide the reasons for dropout, after the participants had enrolled.

We agree with the reviewer and have added this information (lines 103-106).

-how did you define bed-days. In the methods, only incidence rates as 1000 person-years at risk are mentioned, and presumably the time the residents were at risk was only when they were in the care homes, as the authors state they were censored when they left the study care home.

We accept the reviewer's point and have added text to the methods to clarify the definition we used (lines 86-88).

-as in the abstract, the statement 'we observed seven outbreaks in study participants in these care homes' is unclear. Suggest you add more information on these outbreaks—which care homes, how many participants were involved in each outbreak, and how the outbreak cases were similar or different from endemic (sporadic) cases (e.g., symptoms, gender, age, location in facility, etc)

We agree with the reviewer and have added text to address this point (lines 131-136).

-page 7, lines 128-13. The following statement is unclear: "The median time difference for samples 129 positive for norovirus was 0 days (range 0-1 days). This was significantly shorter (Wilcoxon rank sum 130 test, p-value = 0.016) than the difference for samples which were negative (median 4 days, range 1- 131 18 days)" what do the authors mean by time difference? Are they saying that samples that had a faster collection time from the onset of symptoms were more likely to be positive for norovirus? If so, please clarify. Also, this is to be expected as norovirus is most likely to be detected in the first few days after onset of illness, even if it can shed in the stool for weeks following the initial infection.

We thank the reviewer for highlighting this point. We have amended the text to clarify that we are referring to the delay between symptom onset and stool submission, as suggested by the reviewer (lines 146-149). We have also added text to more explicitly refer to this in the discussion (lines 184-185).

Discussion

-first paragraph comparing incidence rate in the current study (76.4 outbreaks per 100 care homes per year) with a previous study (37.1 outbreaks/100 care homes/year)—the authors have not included any confidence intervals or measures of uncertainty, so the reader cannot assess if this is a true difference.

We thank the author for making this point. We have included a confidence interval around this estimate (lines 132-133) in the results and incorporated this into the discussion (lines 157-158).

-first mention that 89% of cases were part of an outbreak. Where are these data in the results? Again, more of a description on the outbreaks (e.g., number of cases in each, facility control measures, etc) and outbreak cases is needed to understand their importance.

We accept that this information should have been included in the result section. We have now added it (lines 136-137), along with more information on the outbreaks (lines 133-136).

-page 10, line 252: how did the authors calculate the following data point: “this study implies that the total burden will be underestimated by around 25%.” What do the authors mean by ‘burden’ here—clearly, it is not the % of cases included in outbreaks out of the total, which was previously stated in the discussion as 89%. Additionally, if this is the main finding of the study, warranting inclusion in the concluding paragraph as well as the abstract, it needs to be clear to the reader from the results and prior discussion how this was calculated.

We take the reviewer’s point, along with the comment regarding the abstract and have reworded this section to more clearly follow the results presented in this manuscript (lines 270-275).

-the majority of the discussion focuses on outlining the strengths and weaknesses of the study, rather than orienting the reader to the importance of the study findings.

We appreciate the view of the reviewer that we have a good amount of discussion on the strengths and weaknesses of the study, however we disagree that there is insufficient discussion of the importance of the study findings. We use 4 paragraphs to highlight the importance of the study findings (lines 155-188) and place these results within the context of international literature (lines 257-267).

Reviewer: 2

Reviewer Name: Carly Adams

Institution and Country: Emory University Rollins School of Public Health, USA Please state any competing interests or state ‘None declared’: None declared

The methods described will sufficiently allow the study to be repeated once authors elaborate on how and when participants were recruited (see attached).

This paper presents results from a prospective cohort study examining the incidence of individual gastroenteritis cases in care homes in North West England, United Kingdom. Authors used active surveillance data to calculate the incidence rate of gastroenteritis illness in care homes, both overall and stratified by residents and staff. Because studies examining gastroenteritis incidence in this setting are limited, and surveillance systems focus on gastroenteritis outbreaks rather than individual cases, this paper confronts the important issue of estimating the incidence of gastroenteritis in care homes.

Overall, the study is well conducted and uses prospectively collected data, minimizing concerns about reporting bias and/or recall bias. Authors adequately address limitations of the data, noting concerns about generalizability of results to other areas of England and different developed countries and concerns about selection bias due to potential differences between study participants and eligible individuals who did not (or were unable to) consent to take part. However, it is not entirely clear from the paper how PTAR was calculated for participants in the study, leading to concerns about incidence rate calculations. Furthermore, there are discrepancies in the paper for reported incident rates, particularly the incident rate for residents.

Specific comments follow.

Major:

1. Page 4, Lines 39-46: While it states that PTAR commenced upon recruitment into the study, it is unclear when and how participants were recruited. Was there a start date when all participants were recruited, after which new care home residents/staff were recruited upon entering the care home? Was this start date the same for all care homes? I would be particularly concerned if new participants were ascertained because they became ill with gastroenteritis, and were recruited into the study as a result of this illness.

We accept the reviewer's comment that more detail is required around the recruitment into this study. We have added text to clarify these points (lines 44-47).

2. Page 4, Lines 39-46: Were participants screened for gastroenteritis prior to recruitment? I recommend addressing this to ensure prevalent cases were not included in the incidence rate calculations.

We thank the reviewer for raising this issue. Indeed prevalent cases were not included, we have clarified the text to reflect this (lines 46-47).

3. Page 5, Line 94: How was PTAR calculated for staff? Was it calculated the same for staff and residents? Particularly, was the amount of time spent at the care home (i.e., hours worked) taken into account? If not, I recommend addressing this in the limitations.

PTAR was calculated in the same way for staff and residents, we have added text accordingly (lines 88-89). We have not adjusted for time spent by staff in the care home as we are interested in all gastroenteritis cases in this group because we did not feel source attribution would be feasible, therefore we do not believe it is necessary to add this as a limitation.

4. Page 6, Lines 106 and 108; Page 7, Lines 149-150; Page 10, Line 236; Table 2: The incidence rate in care home residents, which is one of your main findings, is not consistent. On page 7, the incidence rate in residents is listed as 0.69 cases per 1000 bed-days. On pages 8 and 11, the incidence rate in residents is listed as 0.78 cases per 1000 bed-days. It is unclear why these are different.

We are grateful to the reviewer for highlighting this point. We have corrected these incidence figures in each of the four places. Please see below for an explanation of why this issue occurred.

Minor:

5. Page 3, Lines 28-29: While participating care homes could not be compared to those that did not participate, it would be useful to have more information about the care homes that did agree to participate. Were they recruited prospectively or reactively? How many other homes were invited to participate but declined? Were they all recruited at the same time?

We agree with the reviewer that it would be useful to have this information and have added text to clarify this point (lines 31-32).

6. Page 5, Lines 39-40: Was this questionnaire administered to all care homes at the beginning of the study period? Was it administered only once? Please clarify.

We have added text to explain that this was collected at the start of the study period (lines 41-42).

7. Page 4, Lines 53-55: How were non-infectious causes determined? Were these determined by a trained nurse as well? By self-report? It would be useful to include the number of cases excluded for this reason.

This was done by a trained research nurse. We have included text to clarify this (line 62) and added to the results that no cases were excluded for this reason (lines 124-125).

8. Page 5, Line 76: Incidence rates were not calculated for norovirus cases only, as only 3 cases tested positive for norovirus. I recommend removing this.

We agree with the reviewer and have removed this from line 84.

9. Page 6, Line 111 (Table 2):

1. Please specify units for the incident rate.

2. Because the incident rate is presented as cases per 1,000 person years, I recommend converting PTAR to years.

3. PTAR for all cases is listed as 82,358 days, however this column sums to 122,898.

4. The number of cases for all residents is listed as 41, however this column sums to 39.

5. The incidence rate for residents is listed as 252.5, however $41/(66,489/365) = 225.1$.

Furthermore, $41/66,489 = 0.62$ per 1000 bed-days, which is different than that presented in the text.

We are extremely grateful to the reviewer for their careful appraisal of this table. We have now corrected the analysis to properly include participants with multiple episodes. This had a small effect on both the numerator and the denominator for the incidence calculations. Incidence rates have been corrected throughout the manuscript but their magnitude is similar and this does not affect the conclusions we draw from these results.

Additionally, we have accepted the reviewer's comments and specified the units for the incidence rates and converted the PTAR column to person-years.

10. Page 7, Line 124: Table 2 lists 4 cases in staff (not 3).

We thank the reviewer for spotting this, this typo has now been corrected.

11. Page 7, Line 145: Another likely reason for this difference is that staff are not exposed as long as residents. Staff will go home after working (i.e., they are only exposed while working at the care home), whereas residents are presumably always being exposed.

We agree with the reviewer and have added this additional explanation (lines 168-169).

12. Page 10, Lines 244-245: The study referenced here estimated prevalence, not incidence, of gastroenteritis in the community. I recommend removing this, as the incidence rate from this study should not be compared to a prevalence estimate.

We take the reviewer's point and have amended this reference to one which estimates incidence of gastroenteritis in the elderly in the community.

VERSION 2 – REVIEW

REVIEWER	Carly Adams Emory University, USA
REVIEW RETURNED	16-Nov-2019

GENERAL COMMENTS	Authors have addressed all of my concerns; I have no further comments.
--